# Sustained-Release Biodegradable Intracameral Implants Containing Dexamethasone and Moxifloxacin: Development and In Vivo Primary Assessment

**DOI:** 10.3390/pharmaceutics17091191

**Published:** 2025-09-13

**Authors:** Pablo Miranda, Luis Ignacio Tártara, Analía Castro, Patricia Zimet, Ricardo Faccio, Santiago Daniel Palma, Álvaro W. Mombrú, Helena Pardo

**Affiliations:** 1Laboratorio de Nanotecnología, Instituto Polo Tecnológico de Pando, Facultad de Química, Universidad de la República, Canelones 91000, Uruguay; pmiranda@fq.edu.uy (P.M.); acastro@fq.edu.uy (A.C.); pzimet@fq.edu.uy (P.Z.); 2Cátedra de Física Biomédica, Facultad de Ciencias Médicas, Universidad Nacional de Córdoba, Córdoba X5016JXA, Argentina; ignacio.tartara@unc.edu.ar; 3Unidad de Investigación y Desarrollo en Tecnología Farmacéutica, Facultad de Ciencias Químicas, Universidad Nacional de Córdoba, Córdoba X5016JXA, Argentina; sdpalma@unc.edu.ar; 4Área Física, Facultad de Química, Universidad de la República, Montevideo 11800, Uruguay; rfaccio@fq.edu.uy (R.F.); amombru@fq.edu.uy (Á.W.M.)

**Keywords:** cataract, endophthalmitis, intraocular implants, dexamethasone, moxifloxacin

## Abstract

**Background/Objectives**: We report the development of a novel intraocular sustained-release implantable pharmaceutical formulation, designed to be placed in the anterior chamber of the eye after cataract surgery. The device is intended to reduce postoperative inflammation, and to prevent opportunistic bacterial infections that may lead to endophthalmitis. **Methods**: The implants were produced via hot-melt extrusion, using a twin-screw extruder to process a homogeneous mixture of polylactide-co-glycolic acid, moxifloxacin hydrochloride (MOX HCl) and dexamethasone (DEX). Quality control tests included drug content determination, release rate profile evaluation, and several instrumental characterization techniques (scanning electron microscopy (SEM), confocal Raman microscopy, differential scanning calorimetry, and X-ray diffraction). Long-term and accelerated stability tests were also performed, following ICH guidelines. Sterilization was achieved by exposing samples to gamma radiation. In vivo exploratory studies were carried out in healthy rabbits to evaluate the safety and overall performance of the implantable formulation. **Results**: In terms of quality control, drug content was found to be homogeneously distributed throughout the implants, and it also met the label claim. In vitro release rate was constant for MOX HCl, but non-linear for DEX, increasing over time. In vivo preliminary tests showed that the inserts completely biodegraded within approximately 20 days. No clinical signs of anterior segment toxic syndrome or statistically significant intraocular pressure differences were found between treatment and control groups. **Conclusions**: The implants developed in this study can act as sustained-release depots for the delivery of both DEX and MOX HCl, and are biocompatible with ocular structures.

## 1. Introduction

The eye is isolated from peripheral circulation by the blood–retinal barrier and the blood–aqueous barrier. The first of these separates capillary vessels from the vitreous cavity, whilst the second separates those vessels from the anterior chamber. This latter anatomical space is located between the cornea and the lens. Circulating within it is the aqueous humor. The main functions related to this fluid are maintaining a physiologically stable intraocular pressure and delivering nutrients to avascular tissues such as the cornea and the lens [1]. Aqueous humor production relies on active secretion from structures called ciliary bodies, which are located in the most anterior and peripheral region of the retina (*ora serrata*). They are also responsible for the lens accommodation process, which facilitates focusing on objects at different distances. However, some problems begin to arise with age. The lens proteins that make up its structure denature over time. An opaque cloud that interferes with the normal path of light is subsequently formed. This results in a condition known as cataract, which is one of the most prevalent ocular pathologies worldwide in aging populations [2]. The treatment consists of surgically removing the lens and replacing it with an artificial substitute [3]. Although this intervention is currently considered routine in the field of ophthalmology, it usually provokes a certain degree of undesired inflammation [4]. More importantly, if proper aseptic measures are not implemented during the procedure, there is a risk of bacterial infections leading to a serious pathology called endophthalmitis [5]. Therefore, the usual post-surgery treatment aims to relieve inflammation symptoms and prevent microorganisms from colonizing internal eye structures. To accomplish this treatment, eye-drops containing corticosteroid [6] and fluoroquinolone antibiotic drugs [7] are generally administered. However, the absorption rate through external eye structures is quite low, partly due to lacrimal drainage. For example, ophthalmic solution formulations typically exhibit bioavailability values under 7% [8]. This means that in order to reach therapeutically adequate intraocular concentrations, it is necessary to perform multiple closely spaced instillations for several days. In aging patients, such a scenario can impact treatment adherence. A viable alternative to this problem could be the introduction of sustained-release biodegradable implantable devices into the anterior chamber during surgery. In this way, patients would consistently receive adequate dosing for several days after the procedure, without the need to worry about adhering to the treatment. Ocular implants technology appeared in the 1970s when the pilocarpine delivery system Ocusert^®^ was marketed as a novel solution for glaucoma treatment. This was the first device of its kind, designed to release the drug after being placed in the conjunctival sac [9]. Several other intraocular systems have since been developed, most focusing on intravitreal drug delivery [10]. Some devices were designed as non-biodegradable matrices (such as Retisert^®^ and Illuvien^®^, both containing fluocinolone acetonide), but this technology was slowly replaced by biodegradable analogs, mainly based on poly-lactic-co-glycolic acid (PLGA). By far, the most commercially successful intraocular device has been Ozurdex^®^ (Allergan, USA). This system was designed to deliver DEX from a PLGA matrix to the vitreous for up to six months [11]. PLGA is a particularly useful polymer because of its biocompatibility and biodegradation, making it a very suitable and versatile component for intraocular drug delivery [12]. Ozurdex^®^, Retisert^®^ and Illuvien^®^ (the most well-known intravitreal devices) were aimed at ailments of the posterior segment. However, there have been few attempts over the years to develop implantable systems for intracameral drug delivery. One of the few examples was Surodex^®^, a formulation consisting of a 1 mm diameter pellet containing dexamethasone (DEX), designed to deliver the drug for up to ten days, and to be placed in the anterior chamber during cataract surgery [13]. Clinical trials showed that this device performed better than eye-drops with the same drug [14]. However, Surodex^®^ was only able to deliver DEX for up to ten days, which might not be enough to guarantee anti-inflammatory benefits during a recovery period that may last several weeks. More recently, focus has been shifted from the vitreous to the anterior chamber, with some companies like Allergan and Glaukos obtaining FDA approval for implantable devices aimed at controlling open-angle glaucoma. Particularly, Allergan is currently marketing Durysta^®^, a PLGA-based bimatoprost (prostaglandin analog) sustained-release device intended for 3 to 4 months of drug delivery [15]. On the other hand, Glaukos is marketing an extended-release non-biodegradable device named iDose^®^ based on an ethinylvinylacetate matrix, delivering travoprost (another prostaglandin analog) for up to six years [16]. Although sophisticated, commercial intraocular implants have traditionally been designed as single-drug vehicles aimed at treating a certain pathology or group of related pathologies. This is especially true for inflammatory processes, in which a corticosteroid drug is intended to control a wide variety of conditions, such as uveitis, diabetic macular edema, etc. To the best of our knowledge, there is no commercially available intraocular formulation containing more than one drug to treat completely unrelated conditions. Considering the promising precedent of Surodex^®^ and the current trend in pharmaceutical technology focusing on anterior chamber sustained-release devices, this research work focused on developing, characterizing and evaluating the overall behavior of a novel intracameral implantable formulation based on PLGA containing DEX, while simultaneously incorporating moxifloxacin hydrochloride (MOX HCl).

## 2. Materials and Methods

### 2.1. Chemicals

DEX with 99.20% purity and a particle size distribution of 90% < 30 µm by volume was supplied by Zhejiang Xianyu Pharmaceutical (Taizhou, Zhejiang, China). MOX HCl with 99.90% purity and a particle size distribution of 90% < 68.20 µm by volume was obtained from Shreeji Pharma International (Vadodara, Gujarat, India). Poly-lactide-co-glycolic acid (Resomer RG502H) with a molecular weight of 7000–17,000 Da, a viscosity range of 0.16–0.24 dL/g and a copolymer ratio of 48–52% was purchased from Evonik Industries (Darmstadt, Germany). All other reagents used in this research were of analytical grade.

### 2.2. Reverse Phase High-Performance Liquid Chromatography (RP-HPLC) Determination

An isocratic RP-HPLC method [17] was modified and validated to assess the amount of drug present in different powder and implant samples. Mixed standard stock solutions were prepared by dissolving 12.50 mg of DEX and 62.50 mg of MOX HCl in 25 mL of mobile phase (final concentrations of 500 µg/mL and 2500 µg/mL, respectively). The mobile phase consisted of 56% methanol and 44% buffer solution prepared by dissolving 2.72 g of monobasic potassium phosphate in 1 L of purified water and adding 1 mL of triethylamine. The pH of the mixture was adjusted to 2.80 with concentrated phosphoric acid. Measurements were performed on a Dionex Ultimate 3000 HPLC system (ThermoFisher Scientific, Waltham, MA, USA), using a 250 × 4.60 mm, 5 µm particle size BDS Hypersil C8 column (ThermoFisher Scientific). Total runtime was 10 min, detection wavelength was 242 nm, flow rate was 1.5 mL/min, temperature was 30 °C and injection volume was set to 20 µL. This ensured accuracy and reproducibility for subsequent analyses of drug content and release rate.

### 2.3. Particle Size Reduction

This procedure was only performed on DEX since its low solubility had to be compensated for in order to achieve an adequate release rate suitable for our needs. A total of 50 g of DEX was placed in a custom-built mostly cylindrical steel container (estimated volume of 1.70 L), designed to fit properly in a Turbula T2F Shaker Mixer (Willy A. Bachofen, Muttenz, Switzerland). A sufficient number of stainless-steel spheres of different sizes was added, and the equipment was set to a rotation speed of 49 rpm. We used a total of 10 spheres with a diameter of 20 mm, and 70 spheres with a diameter of 10 mm. The micronized powder was removed from the container after three hours, and then passed through a 30-mesh sieve. The particle size distribution was measured using an LS30 static light particle analyzer (Beckman Coulter Inc., Brea, CA, USA). The resulting profile was consistent with the requirements for sustained intraocular drug delivery over a period of several days. We did not micronize MOX HCl because its high solubility in water would have led to a very quick implant disintegration. In the case of PLGA, the reason was its elevated cost as well as the tendency for material losses during the process.

### 2.4. Powder Homogeneity Determination

This study aimed to quantify the percentage of drug in the powder mixture. Five samples weighing 10 mg were placed in 10 mL volumetric flasks, and 5 mL of acetone was added to dissolve DEX and PLGA. Mobile phase was then added, for a total of 10 mL. This step allows the dissolution of MOX HCl and the selective precipitation of the PLGA polymer. A total of 100 µL of these solutions was then mixed with 150 µL of acetone and diluted to 2 mL with mobile phase (final theoretical concentrations of 50 µg/mL for DEX and 250 µg/mL for MOX HCl, respectively). Finally, 1 mL of each sample was filtered through 0.45 µm polyamide membranes before injection into the HPLC system. The homogeneity was then calculated by comparing the experimental amount of drug determined and the theoretical amount, assuming perfect mixing conditions. For each experiment, samples were taken from each of the four corners and the center of the container. Determining homogeneity ensured that the starting material for implant production already met good quality standards, so that subsequent processing would deliver an even better product.

### 2.5. Optimal Mixing Time Determination

To achieve uniform distribution of both drugs within the polymer matrix, we evaluated the mixing process at different durations. A total of 25 g of MOX HCl, 20 g of RG502H and 5 g of micronized DEX were placed in a plastic container designed to fit the Turbula T2F Shaker Mixer. All components were added sequentially in that specific order. The powder was then mixed at a rotation speed of 72 rpm for 60, 90, 120 150, or 180 min. The optimal mixing time was defined as the point at which the amount of both drugs in the mixture approached their corresponding theoretical percentages with the lowest possible relative standard deviation (RSD) [18]. This step identified the conditions that yielded the most homogeneous powder blend, ensuring implant content reproducibility.

### 2.6. Implant Production

The inserts were produced via hot-melt extrusion, a technique chosen for its ability to ensure homogeneous drug distribution within biodegradable polymers such as PLGA. A total of 50 g of the mixed powder was placed in a DDW-MT volumetric hopper (Brabender, Duisburg, Germany), feeding a Pharma 11 Twin Screw Extruder (ThermoFisher Scientific) continuously refrigerated with deionized water via an Accel Laboratory Series water recirculator (ThermoFisher Scientific) and equipped with a 0.50 mm die. All five stations of the extruder were set to 150 °C, and the rotation speed of the screws was set to 120 rpm. These conditions were found to be suitable for obtaining units with good homogeneity and release rate, according to preliminary runs. The production was performed in a Class C room, under laminar flow. The yellow fiber produced was sealed in a waterproof plastic bag and later manually cut into cylindrical implants of approximately 4 mm in length and 0.60 mm in diameter. The selected dimensions were chosen based on a consensus between the maximum tolerable size for intracameral placement (according to an expert ophthalmologist) and the minimum volume to achieve the desired doses. Inserts of 1.90 to 2.10 mg were then accurately weighed using a Shimadzu AUX 220 analytical scale and sterilized with gamma radiation at a dose of 30 kGy in a Gammacell 220 Excel system (Best Theratonics, Ottawa, ON, Canada). This procedure allowed us to obtain uniform inserts of defined dimensions and drug load, suitable for quality control and in vivo testing.

### 2.7. Uniformity of Content

The standard procedure is described in United States Pharmacopeia (USP 42–NF 37); Chapter <905> Uniformity of Dosage Units. United States Pharmacopeial Convention: Rockville, MD, USA, 2019. Since the inserts of this work deliver less than 25 mg of drug, the uniformity of content test was used. Briefly, the drug content for 10 individual units was determined, and the average compared as a percentage to the label claim (200 µg for DEX and 1000 µg for MOX HCl). Then, the accepted value was calculated as is the 95% significance statistical coverage factor, s is the standard deviation of the individual drug contents, $\bar{P}$ is the average percentage of drug in individual units and *M* depends on the value of $\bar{P}$. *M* = 98.50% if $\bar{P}$ < 98.50%, *M* = 101.50% if $\bar{P}$ > 101.50 and *M* = $\bar{P}$ if $\bar{P}$ is between 98.50% and 101.50%.(1)$$\mathrm{AV}=\left|M-\bar{P}\right|+\mathrm{Ks},\;\mathrm{with}\;K=2.40$$

The test is passed if AV for the batch is ≤ 15. If AV > 15, 20 additional units are analyzed. The batch is accepted in this case if AV ≤ 25 and each individual percentage is between 0.75 M and 1.25 M, respectively. Drug content analysis was carried out by first disintegrating each individual insert in 400 µL of acetone. In a following step, 1.60 mL of mobile phase was added. After 30 s of vortex stirring, 1 mL of the suspension was diluted to 2 mL with mobile phase. All samples were then filtered through 0.45 µm polyamide membranes and the solutions were injected into the HPLC system.

### 2.8. Drug Release Study

To evaluate the ability of the implants to provide sustained delivery, we performed in vitro release assays under sink conditions to reduce saturation issues. This was defined as the condition in which the drug concentration never exceeds 20% of their corresponding solubility limit [19]. A total of five experiments were carried out, using a 0.90% m/V sodium chloride solution as the dissolution medium. Five implants of known mass and drug load were separately placed in 10 mL volumetric flasks. Then, 0.90% m/V saline solution was added to complete a total of 10 mL. After that, the flasks were placed in an ES-20 Orbital Shaker Incubator (Biosan, Riga, Latvia) set to a constant temperature of 37 °C and 140 rpm stirring. A total of 1 mL of sample from each flask was taken with replacement on days 1, 2, 3, 7, and 10, respectively. All samples were then filtered through 0.45 µm polyamide membranes, injected into the HPLC system and compared to a 1/50 dilution of the mixed standard stock solution. This study was carried out before and after sterilization to assess possible performance changes. It also established the baseline release profiles for subsequent comparisons.

### 2.9. Stability Testing

To assess the long-term reliability of the formulation, stability studies were carried out under both accelerated and recommended storage conditions. This study provides information about the correct conservation method for the implants. PLGA has a glass transition temperature of around 42 °C. However, the glass transition can slowly start even at room temperature. In addition, PLGA is a highly hygroscopic polymer. Therefore, the implants were intended to be stored in vacuum-sealed waterproof containers, in a refrigerator set between 4 and 8 °C. This would prevent both problems. According to ICH guidelines, accelerated stability studies must be carried out at 25 ± 2 °C for 180 days for these kinds of storage conditions. A controlled environment chamber (Binder GmbH, Tuttlingen, Germany) was used to ensure such conditions remained constant for six months. Long-term stability studies, on the other hand, should be performed for at least 1 year at 5 ± 3 °C. In this case, an ordinary refrigerator was used. Both for accelerated and long-term stability testing, the recommended time interval between sample analyses is once every 3 months. In this work, we determined drug content, the amount of degradation products and the full 10-day in vitro release profile at months 3, 6 (accelerated/long-term), 9, and 12 (long-term only), and compared the results to those obtained for a reference batch at time = 0. ANOVA statistical analysis was also performed to determine differences between exposure times. These results were essential to define adequate conservation parameters.

### 2.10. Confocal Raman Microscopy

The homogeneity of the implants was evaluated using confocal Raman spectroscopy with an Alpha 300-RA microscope (WITec GmbH, Ulm, Germany). Raman spectra were acquired for the individual raw materials—DEX, MOX HCl, and PLGA—using a 785 nm excitation laser. Each spectrum was recorded as the average of 100 measurements, with a total integration time of 0.50 s per acquisition. Samples were deposited on a silicon wafer prior to analysis. Spectral data from the raw materials were compared with those obtained from the composite implants to assess potential chemical interactions between the components. This information was processed by the software to generate spatial distribution maps of each compound within the sample. The resulting chemical images were used to qualitatively evaluate the homogeneity of the final formulation.

### 2.11. Differential Scanning Calorimetry

Thermal analysis was performed with a DSC 60A system (Shimadzu, Kyoto, Japan). Samples weighing 5 mg (implants, raw materials) were placed in an open aluminum pan and heated from 30 °C to 300 °C at a rate of 10 °C/min. Thermal events such as glass transitions, melting points or decompositions were determined, and the behavior of each compound qualitatively assessed.

### 2.12. Scanning Electron Microscopy

The surface structure of the implants was studied with a JSM 5900LV SEM system (Jeol, Tokyo, Japan). Samples were previously coated with gold, and analyzed under a voltage of 10 kV.

### 2.13. X-Ray Powder Diffraction

Measurements were performed with a Rigaku Miniflex 600-C X-ray diffractometer (Rigaku, Tokyo, Japan) operating in Θ − 2Θ Bragg–Brentano geometry, equipped with a generator for Cu Kα radiation, λ = 1.5419 Å, operating at 40 kV, 15 mA, and utilizing a D/tex Ultra 1D detector. Studies were carried out in scan mode over a 2Θ range from 5° to 55° with a step size of 0.02° and a scan speed of 10 °/min.

### 2.14. In Vivo Exploratory Studies

To preliminarily assess the safety and behavior of the implants inside the eye, we conducted an exploratory study in rabbits. During a one-week adaptation period, five non-randomized albino female, non-pregnant New Zealand rabbits weighing 2.50 to 3 kg and aged 3 to 8 months were fed ad libitum in a controlled temperature room (21 ± 5 °C), exposed to 12-h light/dark cycles. The animals were not genetically modified. To minimize potential confounders, the rabbits were housed in numbered individual cages with environmental enrichment (wooden bar), in the bioterium of the Facultad de Ciencias Médicas (FCM) of the Universidad Nacional de Córdoba (UNC). In these studies, the number of test animals should be minimized based on 3R (Reduce, Refine, Replacement) principles. Appropriate measures were taken to minimize discomfort and pain in the rabbits. The experiments were carried out following the Institutional Committee for Care and Use of Laboratory Animals guidelines (CICUAL, FCM-UNC), in adherence to the National Institute of Health (NIH) guidelines for the care and use of laboratory animals. The project was approved by CICUAL–FCM–UNC Res. CE-2023-00288395. Before experimentation, the animals remained in their cages for two weeks to acclimatize. All of them were included to start the study. All surgical procedures were performed by a professional ophthalmologist. First, the animals received general anesthesia (intramuscular injection of ketamine 3–5 mg/kg and xylazine 0.5–1 mg/kg). Local anesthesia consisting of a proparacaine ophthalmic solution was also applied. The pupils were then dilated using tropicamide eye-drops, and a blepharostat was put in place. After that, the rabbits received an implant in the anterior chamber of each eye. The control group consisted of the contralateral eyes that did not receive the insert. The cylindrical implants produced via hot-melt extrusion were previously weighed on an analytical scale and then measured with a digital caliper to ensure they had the correct dimensions (4 mm in length and 0.60 mm in diameter). The introduction was performed using microsurgical ophthalmological instruments and an optical microscope. All procedures were carried out in a room equipped for animal surgery. A 1 mm peripheral, tunneled and self-sealing incision (corneal paracentesis) was performed in the corneal limbus. The implant was placed through the paracentesis into the anterior chamber with curved and blunt-edged ophthalmic forceps, and its free movement was verified by irrigating sterile normal saline solution with a blunt cannula. Finally, self-sealing of the incision was confirmed, as well as the correct formation of the anterior chamber containing the freely positioned insert. The evolution of the implants (size, shape, degradation, position, ocular reactions) was documented for 35 days. Periodic follow-up consisted of slit lamp (Huvitz, Gunpo, South Korea) evaluation by an experienced ophthalmologist, documented with photographs, to determine anomalous insert displacement or inflammatory signs in the anterior segment (conjunctival hyperemia, corneal transparency, anterior chamber flare, keratic precipitates, and other signs of anterior segment toxicity). Intraocular pressure (IOP) readings were also taken, in this case using a digital tonometer (Icare, Vantaa, Finland). Fundus imaging by binocular indirect ophthalmoscopy was conducted if alterations in the posterior segment were observed. Finally, all animals received ketamine/xylazine general intramuscular anesthesia, and were sacrificed by carbon dioxide inhalation in a hood. Student’s t-test mean comparisons for IOP were carried out using Infostat 2024, after verifying normality and homoscedasticity assumptions. The observations collected during the study provided in vivo evidence of the implant’s biocompatibility and behavior.

## 3. Results

### 3.1. RP-HPLC Validation

Since one of the main points of this work was quality control, we relied on an HPLC method validated as per ICH guidelines for solutions containing both MOX HCl and DEX simultaneously. Specificity, linearity, analytical limits, accuracy and precision were assessed. Under the selected chromatographic conditions, these drugs appeared at retention times of about 2.70 and 7 min, respectively. Validation was also conducted at two concentration ranges to cover both drug release studies and content determination analysis, according to Table 1.

Specificity (the ability of the method to correctly separate one compound from another) was evaluated by determining whether the resolution between the peaks corresponding to the main degradation products and those of both drugs was higher than 1. This would indicate that an adequate separation was achieved during the analysis. In this case, main degradation products were generated after treating drug samples with acid and basic media, high temperature, oxidative solutions and light exposure, as per ICH guidelines. The conditions were fine-tuned to obtain, whenever possible, degradation rates between 5 and 20%. This range was selected on the basis that the higher the degradation, the less likely that the products would be spontaneously generated during shelf life. Also, if degradation was too low, there would always be a risk of obtaining very low signals for the impurities. Taking all of this into account, these exposure procedures yielded a total of six degradation products, five of them originating from DEX, according to Table 2.

The rest of the validation parameters (linearity, instrumental limits, accuracy, precision) were studied in two different concentration ranges according to Table 1. This was because the assays implemented for drug content and drug release in vitro were designed to measure different concentrations. Linearity was studied by building calibration curves for these ranges. Accuracy was evaluated at the lowest, middle and highest concentration levels by calculating the recovery percentage of a known amount of an already-spiked drug. These three concentration levels were also used to establish the precision of the method, by calculating the RSD value of the same samples measured intra-day and inter-day, respectively. Results in this work were expressed as 100–RSD, so that a value of 100% represented nominal perfect precision. Analytical limits were determined with the signal-to-noise ratio approach. In this way, the detection limit was calculated as the concentration at which the signal-to-noise ratio was 3. Similarly, the quantitation limit was calculated as the concentration at which the signal-to-noise ratio was 10. Table 3 summarizes the results.

### 3.2. Particle Size Reduction

Before the three-hour milling process, static light scattering analysis revealed that 67.73% of DEX particles by volume were less than 20 µm and 36.70% less than 10 µm, respectively. To reduce particle size, 45 g of unmicronized DEX powder was placed in a stainless-steel container (estimated volume of 1.70 L) designed to fit the Turbula T2F mixer. Several stainless-steel spheres of different sizes were then added. The number of spheres used was carefully selected to optimize the quality of the obtained product. If filled with too many spheres, a cushion effect would prevent optimal milling. If too few spheres were added, it would not sufficiently reduce particle size. After three hours of impact milling, the powder was manually removed from the container walls and submitted to static light scattering analysis. A significant change was observed, with 93.02% of particles by volume being less than 20 µm and 72.37% less than 10 µm. Figure 1 depicts the initial distribution, while Figure 2 shows the particle size profile upon completion of the three-hour milling process.

### 3.3. Optimal Mixing Time Determination

Five different time periods were assessed to determine the point at which the lowest possible RSD was observed. This corresponds to the moment when all components in the mixture achieved their respective theoretical ratios. If too little time is used, the mixture remains incomplete, and if too much time is used, the mixture can become heterogeneous again. The result was obtained after quantifying the overall homogeneity of both drugs in the powder mixture, considering the number of samples taken according to the homogeneity study described in Section 2.5. Results for this study are presented in Table 4, where we expressed the average homogeneity of both drugs for each mixing time along with its corresponding confidence interval (directly correlated with RSD reduction).

### 3.4. Uniformity of Content

After carrying out all the HPLC analyses for each of the individual inserts, the results showed an acceptance value of 6.11 for DEX and 13.08 for MOX HCl, respectively. Table 5 presents the results after quantification performed on a total of 10 individual units, according to USP.

### 3.5. Drug Release Study

Release rate in vitro under sink conditions over 10 days was quantified and is presented as percentages in Table 6. Figure 3A and B show the corresponding release curves, with the equations for DEX and MOX HCl being 1.5804x^1.1552^ (*r*^2^ = 0.9981) and 16.534x − 7.5869 (*r*^2^ = 0.9988), respectively.

### 3.6. Stability Testing

In order to determine the possible effects of the storage conditions (accelerated and long-term) on the amount of drug contained in the inserts and on the release performance, two different studies were carried out. One consisted of a 6-month exposure of the implants to a temperature of 25 ± 2 °C (accelerated conditions). The second involved keeping the inserts at 5 ± 3 °C for 365 days. In both cases, samples were taken every three months. For each cumulative release study, five different implants from the same batch were used. All of them started at time t = 0, and were sampled randomly. The same was true for the uniformity of content study, except that in this case the number of implants analyzed was 10 per session. In summary, a certain number of implants started the six- and twelve-month exposure periods under accelerated or normal conditions. Some were sampled at strategic points in time, while others remained until the end. Table 7 shows the baseline values for degradation, uniformity of content and total amount released after ten days in vitro. Table 8 shows the results after 180 days of accelerated conditions, and Table 9 shows the result after 365 days.

### 3.7. Confocal Raman Microscopy

To examine the intermolecular interactions involved in the formation of the insert, Raman spectra were obtained from the individual components as well as from the final formulation after completing the production process. Figure 4 presents the Raman spectra of DEX, MOX HCl, PLGA, and the optimized insert. Based on these spectra, it can be observed that all individual components are present in the implant without any significant shift in the bands compared to those of the components in their isolated state. This finding indicates that the implant is composed of a physical mixture of each individual component, with no evidence of chemical interactions between them. This result represents an important milestone, because one of the key assumptions of the formulation is that both drugs remain unchanged during the hot-melt extrusion process.

To examine the intermolecular interactions in the implant, Raman spectra were obtained from individual components as well as from the combined formulation. Additionally, Raman spectroscopy was employed to assess whether the powder mixture remained homogeneous in the insert after the thermal extrusion process, which involves the melting of PLGA. Figure 5 shows regions of the sample where DEX (blue), MOX HCl (green), and PLGA (red) were identified based on Raman signal emission. A homogeneous distribution of the components can be observed in this figure (mixed colors in the top left), confirming the results obtained from the content uniformity studies.

### 3.8. Differential Scanning Calorimetry

The analysis was carried out to determine if all thermal events corresponding to the raw materials could be identified in the mixture, considering that they might shift due to the presence of the other components. Individual results for PLGA (glass transition), MOX HCl and DEX (melting points) are shown in Table 10, whereas Figure 6 depicts the thermal events in the mixture.

### 3.9. Scanning Electron Microscopy

Images depicting the surface morphology of the implants were taken before and after sterilization by gamma radiation. The objective was to determine whether there were any physical changes during the process. As will be discussed later, the effect of radiation on the surface of the inserts was already noticeable at 75X magnification. This was reflected in the release rate of both drugs, which became significantly increased after irradiation, giving the inserts their final kinetic profile. Pictures were taken at 75X magnification and are shown in Figure 7A,B.

### 3.10. X-Ray Diffraction

X-ray diffraction (XRD) analysis was conducted to evaluate the crystalline structure of the materials and to confirm the chemical identity of the active pharmaceutical ingredients. Pure PLGA exhibited a broad and asymmetric background signal characteristic of amorphous polymers, corroborating its amorphous nature. In contrast, DEX and MOX HCl displayed clearly defined crystalline diffraction peaks, confirming the characteristic diffraction patterns of these substances [20,21,22,23]. The diffraction patterns obtained for the implant formulation appeared as a superposition of the individual diffractograms of PLGA, DEX, and MOX HCl. The presence of diffraction peaks from the crystalline active compounds embedded in the polymeric matrix indicates that these drugs maintained their crystalline form following incorporation into the implants, confirming successful preparation via hot-melt extrusion. This analysis thus provides evidence supporting the chemical identity and crystalline integrity of the incorporated drugs within the amorphous polymeric matrix. To further investigate the stability of the crystalline structure during processing, the crystalline structure of all three standalone components was studied and then compared to that of the implants before and after irradiation. This experiment was performed to determine if peaks were missing after the manufacturing procedure and subsequent sterilization. Standalone diffraction patterns are depicted in Figure 8, whereas Figure 9 shows a comparison between irradiated (red line) and non-irradiated inserts (black line). The comparison revealed that no diffraction peaks disappeared as a result of the irradiation treatment, further confirming the preservation of the crystalline structure of the drugs throughout the complete manufacturing and sterilization processes.

### 3.11. In Vivo Exploratory Studies

Five rabbits received an insert (previously sterilized with a 30 kGy gamma radiation dose) in their anterior chambers in order to determine the general in vivo behavior of the implantable formulation. Figure 10 shows the degradation progress for days 1, 9, and 15, respectively. The inserts exhibited a fairly constant degradation rate up until day 20, as shown in Figure 11. No local or systemic alterations in the cornea, the iris or lens were detected, confirming good biocompatibility.

Another important parameter that was taken into account in the in vivo study was the IOP of the treated eyes. Tonometry measurements were conducted before placing the insert and after each ophthalmological control. Figure 12 shows that there was no statistically significant increase in IOP (*p* = 0.096) during the time the implant was present in the eye and in the days following its complete disintegration. Moreover, normal values for the rabbits were never exceeded.

## 4. Discussion

### 4.1. RP-HPLC Validation

In order to correctly quantify a drug in any pharmaceutical formulation, a validated analytical determination method needs to be available. In this case, we opted for HPLC analysis and conducted a full validation according to ICH guidelines. The results indicate that the proposed method successfully separates both drugs from their main degradation products, while also being accurate, precise and robust. Furthermore, linearity was achieved for both concentration ranges, and analytical limits were sufficiently distant from the lowest levels of the calibration curves constructed.

### 4.2. Particle Size Reduction

The formulation comprised three components, ranked by mass as follows: MOX HCl (50%), PLGA (40%) and DEX (10%), respectively. The inserts were intended to allow for a sustained release of both drugs. However, DEX and MOX HCl are very different drugs in terms of water solubility. DEX is quite insoluble (approximately 100 µg/mL at 20 °C), whereas MOX HCl is highly soluble (around 20 mg/mL at 20 °C). PLGA, in contrast, is highly insoluble in water. In general terms, particle size reduction is associated with an increase in surface area [24], which also enhances the interaction between drug molecules and solvent, thus speeding up dissolution. This behavior needs to be modulated to obtain the correct release profile for both drugs. Considering all of the above, any particle size reduction for MOX HCl could severely compromise the structural integrity of the insert, producing a sudden antibiotic dissolution burst upon contact with water. On the other hand, PLGA was not a candidate for particle size reduction, as its high market price and the amount of mass lost during our impact milling process would significantly increase the cost of each production run. In the case of DEX, it has been reported in scientific literature that particle size distribution is of paramount importance in intraocular formulations [25]. In order to achieve a sustained release for DEX, it is necessary that at least 70% by volume of particles be under 10 µm. Similarly, 90% of the volume should be below 20 µm. According to the results obtained from static light analysis, the three-hour milling process applied successfully achieved the desired goal.

### 4.3. Optimal Mixing Time Determination

As observed in the results, mixing 5 g of DEX required between 90 and 120 min. In particular, the 2-h process showed the least dispersion and homogeneity closest to 100%. At shorter times, the mixing was incomplete, whereas at longer times the mixture became less uniform. For MOX HCl, mixing for two hours added little improvement over 60 or 90 min. The reason for the behavior of both drugs lies in their amounts in the mixture. MOX HCl differs by only 10% in mass with respect to PLGA, the second most abundant component of the mixture. Together, they account for 90% of the total mass. Since they are present in comparable amounts, mixing is relatively straightforward. DEX, however, requires more time because it is the least abundant component.

### 4.4. Unifomity of Content

Both acceptance values are lower than 15, thus complying with the USP uniformity of content analysis. The average amount of DEX quantified was 96.71%, whereas the MOX HCl content was on average 108.74%. This result suggests that after the two-hour mixing process described, PLGA and MOX HCl are mostly occupying the space DEX should settle in, thus hindering the 100% homogeneity of the inserts. However, the result is very good, considering that there are three components in total, one of them present in a much smaller amount compared to the other two. It should be noted, however, that in every single determination assay, a degradation product appeared in the inserts. The compound was identified with mass spectrometry (data not shown) as 17-oxodexamethasone, which was probably generated during the extrusion process. The Ozurdex^®^ patent reports a maximum tolerance of 1.50% (calculated by comparing peak area of the product to that of the drug). The total amounts determined for our implants were lower than 1.50%. Since this value is within tolerable limits, we expect it to have a negligible impact on the overall safety and performance of the inserts. This was consistent with the fact that no clinical toxicity was observed in vivo.

### 4.5. Drug Release Study

From an in vitro perspective, our implant exhibits a similar or slightly slower release rate than Surodex^®^, making it suitable as a replacement for long-term treatments with eye-drops. Such behavior promises to be highly beneficial in current clinical practice. The amount of drug delivered after 10 days demonstrates that the implant can be considered in vitro as a sustained-release formulation for both drugs. Although the experiment was conducted under sink conditions, MOX HCl displayed a linear curve after fitting a Higuchi model, suggesting Fickian diffusion. In this scenario, the amount of mass released is directly proportional to the square root of time. DEX, on the other hand, displayed non-linear behavior despite the sink conditions of the experiment. This means that the release rate was not constant at all. In fact, it progressively increased over time, fitting a power law Korsmeyer–Peppas non-Fickian model [26]. This behavior arises when more than one type of transport phenomenon is involved in drug release. Since DEX is a minor component of the formulation, it requires dissolution of the other major constituents to increase its exposure to the solvent. The slow release rate suggests that PLGA degradation and subsequent multiple pore formation could be the most important factor behind this behavior. Moreover, in PLGA-based systems, hydrolytic chain scission progressively increases water uptake and porosity, facilitating drug release over time [12,21]. To summarize, the release behavior observed can be explained by the interplay of the three components of the formulation. PLGA undergoes bulk erosion by hydrolysis, generating pores and increasing water uptake of the matrix. This accelerates drug diffusion. MOX HCl is not so affected by this, because of its more ubiquitous distribution in the insert and its very high solubility in water. Regarding DEX behavior, the situation is more complex. Being the least abundant component of the formulation and also poorly water-soluble, it depends more on polymer degradation and steric factors for sustained release. The reduction in DEX particle size prior to extrusion was therefore essential to increase its surface area and dissolution rate.

### 4.6. Stability Testing

In the long-term stability study, samples were stored in a refrigerator at 5 °C, and analyzed every three months. In terms of content uniformity, there were no statistically significant differences. This was not the case for the in vitro drug release performance. The results suggest that under accelerated conditions, the release rate tends to increase markedly for DEX, and just slightly for MOX HCl. This could be the result of the effect of temperature-induced hydrolysis of PLGA chains, which in turn increases porosity and compromises the integrity of the matrix. This would allow both drugs to permeate faster and more freely, particularly DEX, for which kinetics are matrix-dependent. It should also be noted that in this case, the conditions were so harsh for the formulation that after 180 days of exposure, the inserts had already lost their compact form, possibly because the PLGA matrix was fairly degraded. Even if the amount of drug did not change, it would have been pointless to determine it, since at that point the dosage form was essentially lost. With respect to the long-term study, there was no relevant increase in release rate after 12 months of normal storage. Drug content and degradation remained unchanged within the intrinsic analytical error.

### 4.7. Confocal Raman Microscopy

Homogeneity is a very important parameter to monitor, since an adequate drug release depends on how intimately the components are mixed with each other. This is essential to ensure that the amount delivered can be adequately controlled over time. Figure 5 qualitatively depicts how well the mixing was performed. As seen in the image at 10 µm scale, all components appear to be randomly distributed, with a few spots where MOX HCl is not perfectly mixed with the rest of the material. This is probably the reason why during the uniformity of content evaluation, amounts were usually determined between 100 and 110% for MOX HCl, and 95 and 100% for DEX.

### 4.8. Differential Scanning Calorimetry

It can be seen from Table 4 that all thermal events for all compounds shifted to lower temperatures after mixing for two hours. This result indicates successful mixing. First, there is a dilution effect in mixtures, which causes thermal events to become less pronounced, shifting fusion or glass transition temperatures to lower values, since less collective energy is required for the event to occur. It should also be taken into account that when the components in a mixture are in intimate physical contact, heat transfer between them becomes more efficient. As a result, the energy required to reach a thermal event (glass transition, melting) is more evenly distributed throughout the mixture, decreasing the onset temperature. These phase transitions usually span a temperature range, creating microenvironments that modify thermal properties, in which variations in heat flow are mainly caused by the amounts of the coexisting phases in the overall mixture.

### 4.9. Scanning Electron Microscopy

As seen in Figure 5, there is a clear difference in morphology between inserts before and after sterilization. The surface of non-sterilized implants appears relatively smooth at 75X magnification, while that of the sterilized units looks much rougher in comparison. This effect might be caused by a rearrangement of the PLGA structure. It is also possible that this matrix distortion is one of the reasons why an increased release rate for both drugs is consistently observed after sterilization.

### 4.10. X-Ray Powder Diffraction

As seen from the standalone patterns, both DEX and MOX HCl display well-defined peaks, indicating a crystalline solid structure. In contrast, the graph corresponding to PLGA shows a broad envelope profile, which is typical of amorphous materials. When combined in the insert, the signature of each individual component remains noticeable. Peaks from both MOX HCl and DEX can be observed in the implant, along with a central bump corresponding to the amorphous contribution of PLGA. More importantly, the sterilization process had no significant impact on the position or presence of the peaks, indicating that it does not affect the crystalline structure. This is particularly important, since it implies that no new polymorphs were formed during the extrusion process.

### 4.11. In Vivo Exploratory Studies

The first animal trials allowed us to qualitatively determine, over a 35-day period, the behavior of the implants and the respective ocular tolerance. Five rabbits received an irradiated insert in one eye, while the other one was used as a control. The implants were placed in the anterior chamber without difficulty, and we confirmed that their position after insertion was not significantly modified over time. The residence time in the eyes was approximately 20 days, as shown in Figure 10. This was consistent with our findings on drug release profiles, which indicated that a considerable fraction of both drugs was still available for delivery after 10 days of exposure to the release medium. We estimate that under in vitro conditions, about 25% of the implant’s mass (as DEX and MOX HCl) is released to the saline solution. However, it was clearly visible that in vivo conditions accelerated the release of both drugs, since the insert disappeared completely after 20 days. In vitro degradation mechanisms and release rate kinetics have already been discussed in other sections. Additional factors may also play a role, such as enzymatic activity in the aqueous humor and the dynamic mass exchanges occurring in the anterior chamber. Together, these factors make the insert last about 20 days. Such behavior is expected to beneficially impact patient recovery, maintaining relatively high intraocular concentrations of DEX and MOX HCl for a sufficiently long period to both improve recovery outcomes and prevent opportunistic infections. Moreover, MOX HCl and DEX are considered to be very safe intraocular drugs even at high doses [27,28]. Consequently, no anterior segment toxicity was detected, as confirmed by the absence of acute inflammation and corneal edema. This was reported after clinical evaluation by an expert ophthalmologist. It is also worth noting that none of the eyes developed infections. In addition, IOP did not rise significantly. However, it was observed that the implants tended to adhere to the iris, but not to the corneal endothelium. This was likely because the implants maintained closer contact with the iris in their final position. It is important to note that this was influenced by the presence of the lens, which is large in these animals and restricts the size of the anterior chamber. Therefore, in eyes without cataract surgery, the smaller size of the anterior chamber would promote adhesion to the iris.

## 5. Conclusions

We successfully developed an intraocular sustained-release PLGA insert containing DEX and MOX HCl, which was a particularly complex challenge due to the different solubilities of the two drugs. The implants complied with quality control requirements and proved to be stable under refrigerated conditions for at least 12 months. The accelerated study revealed that storage at low temperatures is of paramount importance. Physicochemical characterization demonstrated that the inserts were homogeneously prepared and retained both drugs after the manufacturing process. Moreover, final sterilization with 30 kGy gamma irradiation did not have a significant effect on composition or performance, although it appeared to increase the amount of DEX delivered in the in vitro assays. The preliminary in vivo studies demonstrated that the insert can be easily introduced into the eye, is non-toxic, and remains in place for at least 20 days while biodegrading. From a clinical perspective, the proposed intracameral implant offers the potential to simplify postoperative management of cataract surgery by combining anti-inflammatory and antimicrobial therapy in a single, intraoperative administration. This approach could improve patient adherence and therapeutic outcomes in populations requiring constant care and surveillance, such as the elderly. Nevertheless, some challenges remain, including improvements to the implantation technique and the possibility of implant adhesion or migration within the anterior chamber. Studies in eyes undergoing cataract surgery are required to evaluate these issues, as well as inter-patient variability in aqueous humor dynamics and degradation rates. We intend to continue addressing these aspects through further preclinical studies, including a larger number of eyes.

## Figures and Tables

**Figure 1 pharmaceutics-17-01191-f001:**
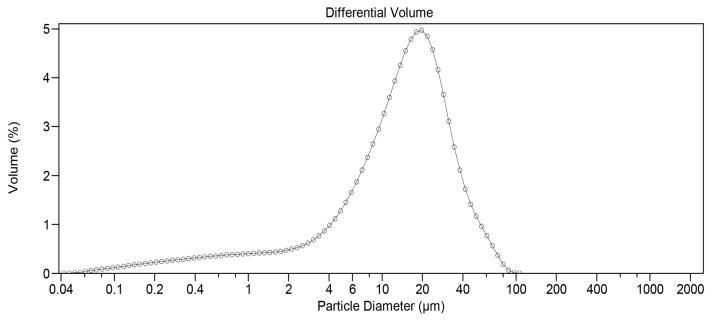
Particle size distribution for DEX before milling (measured by static light scattering).

**Figure 2 pharmaceutics-17-01191-f002:**
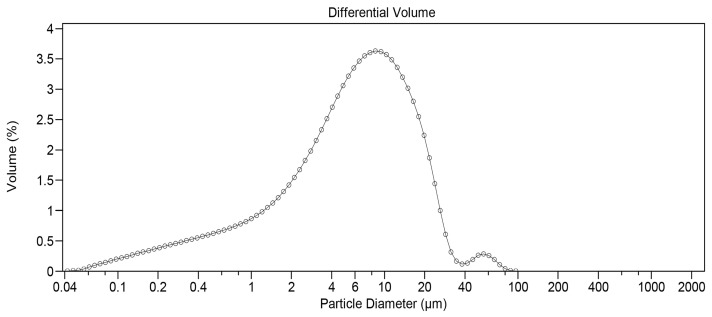
Particle size distribution for DEX after milling (measured by static light scattering).

**Figure 3 pharmaceutics-17-01191-f003:**
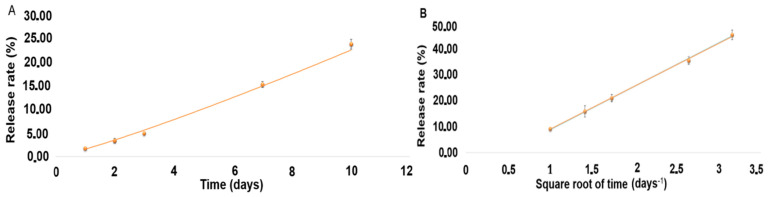
(**A**). Drug release curve for DEX. (**B**). Drug release curve for MOX HCl.

**Figure 4 pharmaceutics-17-01191-f004:**
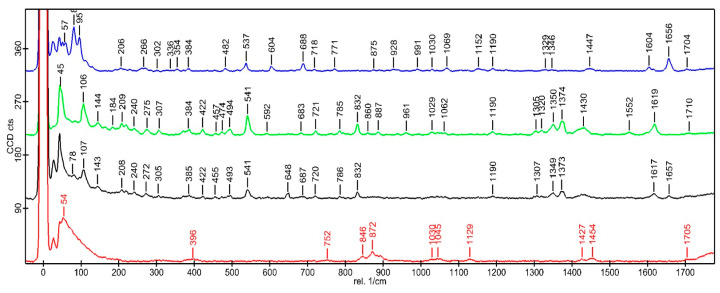
Raman spectra at 785 nm show the characteristic peaks of DEX (blue), MOX HCl (green), insert (black), and PLGA (red).

**Figure 5 pharmaceutics-17-01191-f005:**
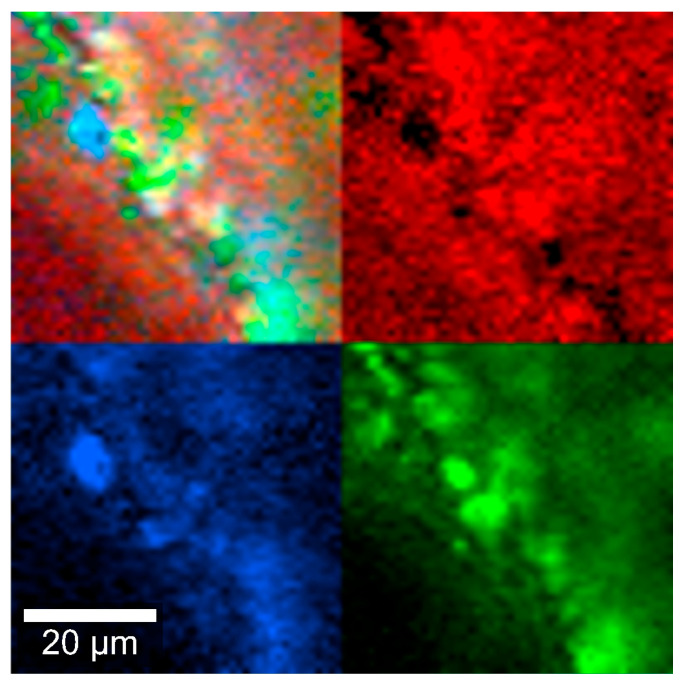
Insert Raman spectra at 785 nm, depicting DEX (blue), MOX HCl (green) and PLGA (red).

**Figure 6 pharmaceutics-17-01191-f006:**
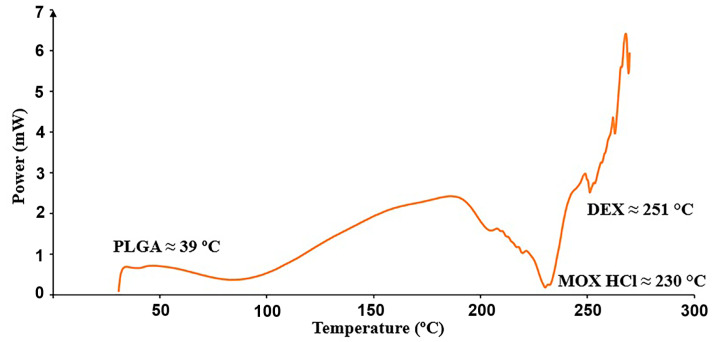
Relevant thermal transitions detected in the insert from 30 to 300 °C at 10 °C/min.

**Figure 7 pharmaceutics-17-01191-f007:**
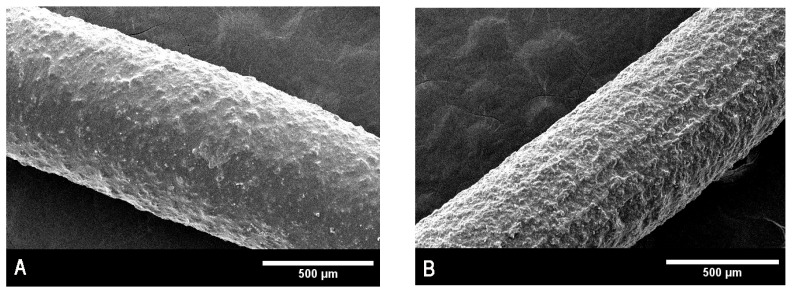
(**A**)**.** Scanning electron microscopy for non-sterilized inserts. (**B**). Scanning electron microscopy for sterilized inserts.

**Figure 8 pharmaceutics-17-01191-f008:**
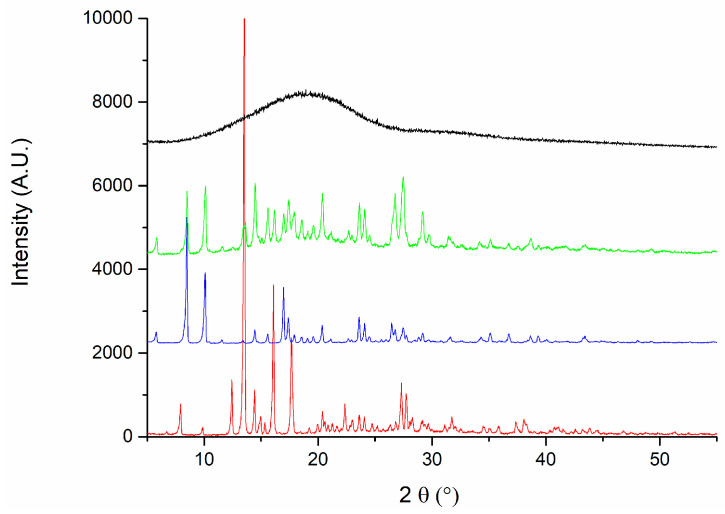
Diffraction patterns for PLGA (black), implants (green), MOX HCl (blue) and DEX (red).

**Figure 9 pharmaceutics-17-01191-f009:**
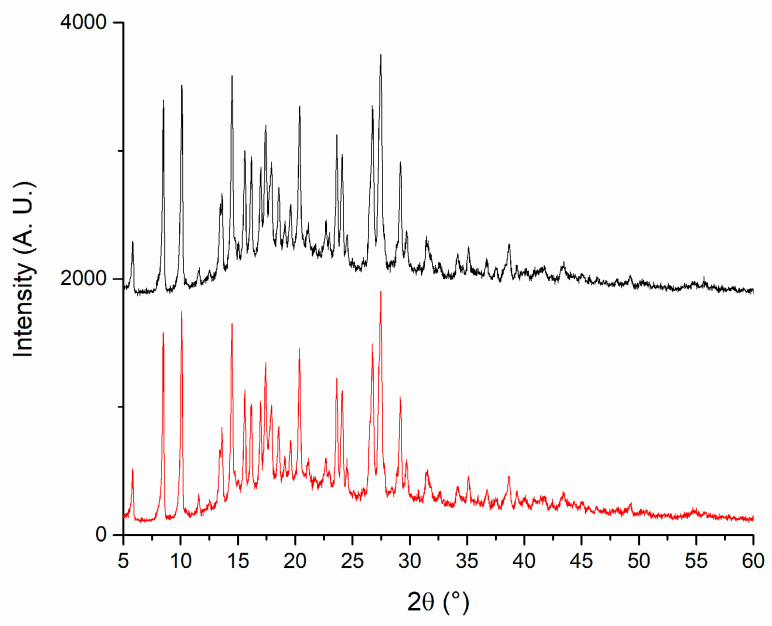
X-ray diffraction patterns for irradiated (red) and non-irradiated implants (black).

**Figure 10 pharmaceutics-17-01191-f010:**
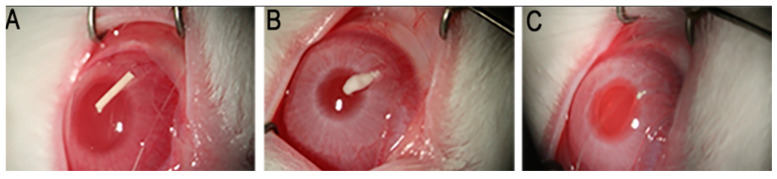
Implant morphology at days 1 (**A**), 9 (**B**) and 15 (**C**) after insertion.

**Figure 11 pharmaceutics-17-01191-f011:**
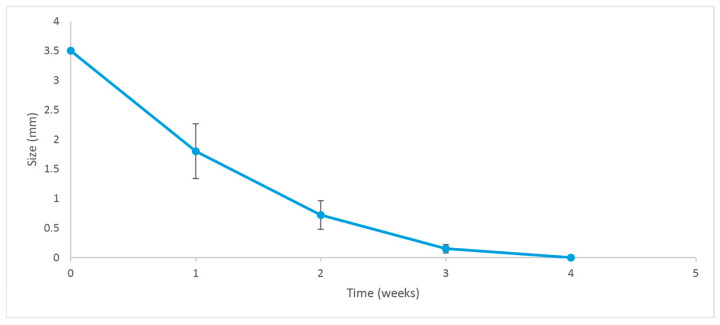
Degradation rate of the implant. Error bars depict the standard deviation of the observations.

**Figure 12 pharmaceutics-17-01191-f012:**
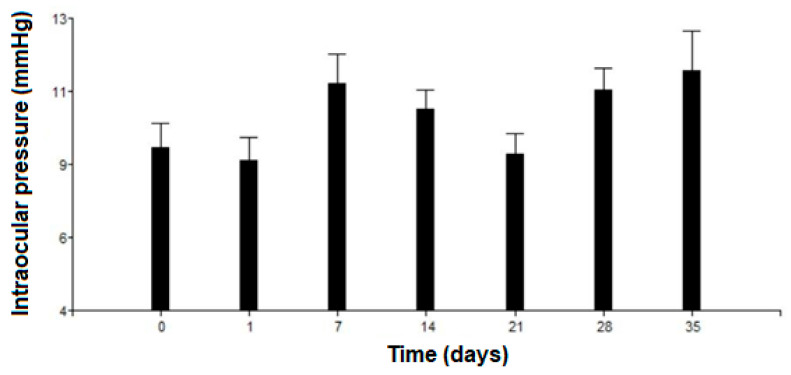
Intraocular pressure variation. Error bars depict the standard deviation of the observations.

**Table 1 pharmaceutics-17-01191-t001:** Concentration ranges used in HPLC method validation.

Low Concentration Range	High Concentration Range
DEX (1–10 µg/mL)	DEX (10–100 µg/mL)
MOX HCl (5–50 µg/mL)	MOX HCl (50–500 µg/mL)

**Table 2 pharmaceutics-17-01191-t002:** Degradation products obtained during validation.

Condition	Degradation Product	Resolution
Acidic medium	DEX-DP1	8.09
Basic medium	DEX-DP2	20.97
Basic medium	DEX-DP3	3.70
Basic medium	DEX-DP4	2.00
Oxidative stress	DEX-DP5	3.51
Thermal stress	MOX-DP1	2.70

**Table 3 pharmaceutics-17-01191-t003:** Analytical validation parameters.

Parameter	DEX	MOX HCl	Range
Linearity	0.9991	0.9990	Low
	0.9999	0.9999	High
Detection limit (µg/mL)	0.09	0.05	Low
	0.09	0.07	High
Quantification limit (µg/mL)	0.29	0.17	Low
	0.27	0.23	High
Accuracy (%)	99.44	99.75	Low
	100.43	101.14	High
Precision (%)	99.17	99.40	Low
	99.67	98.72	High

**Table 4 pharmaceutics-17-01191-t004:** Evaluated mixing times.

Mixing Time (min)	DEX (%)	MOX HCl (%)
60	86.81 ± 13.45	98.81 ± 3.78
90	98.11 ± 2.87	98.45 ± 2.97
120	99.41 ± 2.76	99.17 ± 2.78
150	95.11 ± 7.32	94.14 ± 7.65
180	90.47 ± 3.42	89.08 ± 3.88

**Table 5 pharmaceutics-17-01191-t005:** Uniformity of content study.

Implant ID Number	DEX	MOX HCl
1	95.69 ± 2.02	104.41 ± 2.42
2	95.27 ± 2.03	106.42 ± 2.47
3	95.37 ± 2.03	108.00 ± 2.49
4	96.40 ± 2.08	109.27 ± 2.57
5	99.76 ± 2.08	112.21 ± 2.54
6	97.14 ± 2.10	111.36 ± 2.60
7	93.85 ± 1.99	106.69 ± 2.45
8	97.36 ± 2.07	109.10 ± 2.53
9	97.13 ± 2.09	109.00 ± 2.56
10	99.14 ± 2.09	110.98 ± 2.55

**Table 6 pharmaceutics-17-01191-t006:** Drug release study.

Time (days)	DEX	MOX HCl
1	1.73 ± 0.49	9.15 ± 2.21
2	3.37 ± 0.35	15.86 ± 1.44
3	4.96 ± 0.61	20.96 ± 1.53
7	15.32 ± 1.04	35.36 ± 1.86
10	23.85 ± 1.76	45.32 ± 1.67

**Table 7 pharmaceutics-17-01191-t007:** Baseline values before stability testing.

Study	DEX	MOX HCl
Degradation (%)	1.44 ± 0.07	-
Uniformity of content (%)	96.70 ± 1.05	108.71 ± 1.28
Total amount released (%)	23.85 ± 1.76	45.32 ± 1.67

**Table 8 pharmaceutics-17-01191-t008:** Results after 180 days in accelerated conditions.

Study	DEX	MOX HCl
Final degradation amount (%)	1.36 ± 0.07	-
Final uniformity of content (%)	94.24 ± 1.04	103.68 ± 1.12
Amount released at 90 days (%)	34.25 ± 3.40	53.71 ± 3.18
Amount released at 180 days (%)	-	-

**Table 9 pharmaceutics-17-01191-t009:** Results after 365 days in long-term natural conditions.

Study	DEX	MOX HCl
Final degradation amount (%)	1.39 ± 0.07	-
Final uniformity of content (%)	97.19 ± 1.15	106.62 ± 1.55
Amount released at 90 days (%)	28.78 ± 5.71	47.35 ± 2.97
Amount released at 180 days (%)	25.30 ± 3.34	44.04 ± 2.49
Amount released at 270 days (%)	30.85 ± 3.34	48.96 ± 2.64
Amount released at 365 days (%)	24.21 ± 2.70	42.21 ± 4.85

**Table 10 pharmaceutics-17-01191-t010:** Results after 365 days in long-term natural conditions.

Component.	Melting Point as Raw Material (°C)	Melting Point in Powder Mixture (°C)
PLGA	42–43	≈39
MOX HCl	238–242	≈230
DEX	262–264	≈251

## Data Availability

Raw data can be requested via e-mail to the corresponding author.

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
