# Peer review of "Sustained-Release Biodegradable Intracameral Implants Containing Dexamethasone and Moxifloxacin: Development and In Vivo Primary Assessment"

_pharmaceutics, 2025, doi:10.3390/pharmaceutics17091191_

Round 1

Reviewer 1 Report

Comments and Suggestions for Authors

This study reports the development of a novel intraocular implant designed for sustained release of anti-inflammatory and antibacterial drugs (dexamethasone and moxifloxacin) following cataract surgery. The researchers established a manufacturing process, conducted quality control, stability testing, and sterilization, and performed preliminary in vivo tests in rabbits. The authors achieved important and interesting results by successfully developing a sustained-release PLGA implant containing both dexamethasone and moxifloxacin despite their differing solubilities.

This is a well-structured and high-quality study with thorough work and a carefully characterized system, which makes the paper strong overall. The article is both informative and enjoyable to peruse; however, there are several points that could further strengthen the work and increase its impact:

  1. In the introduction, the authors briefly mention similar developments in this area, but it would be beneficial to provide a more comprehensive overview of this field, including existing limitations of comparable implants and how this study addresses those challenges. Including references to recent studies from the last five years would also enhance context.

  2. The figure captions could be more precise regarding the evaluation methods, especially for Figures 2, 5, and 6. Additionally, clarifying methodologies within the text - for example, specifying that the quantification of DEX and MOX HCl release over 10 days was done in vitro as stated in Table 6 - would improve clarity, even if somewhat redundant given the detailed methodology section. For Figure 5, adding method details and decoding channel information in the caption would make the article more accessible and easier to follow.

  3. A more careful presentation of figures is recommended: Figures 3 and 6 would benefit from sharper fonts, as in low quality of PDF it is hard to read them; Figure 7 should include labels "A" and "B" on images, images better to be separated by a small gap; and Figure 7 requires a clearer scale bar. Also, ensuring consistent font styles and sizes across all figures would improve visual coherence. While not mandatory for publication nor affecting scientific value, these improvements would enhance the article’s appeal to future readers.

  4. Although the methods section is clearly structured and sequential, adding introductory and concluding sentences to sections or paragraphs would help readers better understand the authors’ goals, findings, and the significance of results—especially for those less familiar with the topic. The implant synthesis procedure remains somewhat unclear, so a more detailed description is needed, as the current methodology section does not fully explain the process.

  1. It would be beneficial to include more SEM images at different magnifications and at various stages of the synthesis process to provide a clearer understanding of the implant development.

  2. If possible, comparing the release rates of similar intraocular delivery systems and discussing what release rate is considered optimal or required would add valuable context.

    7. How closely do the in vitro release conditions mimic intracameral conditions? Could the protein composition and chemical microenvironment in vivo significantly alter the drug release rate? Also, how quickly does the released drug diffuse under intracameral conditions?

These comments mainly relate to improving the clarity of the article, better emphasizing its novelty and strengths compared to global analogs, and deepening the understanding of the processes involved. However, they do not diminish the overall scientific significance and quality of the work, and publication is recommended after addressing these points.

Author Response

  1. In the introduction, the authors briefly mention similar developments in this area, but it would be beneficial to provide a more comprehensive overview of this field, including existing limitations of comparable implants and how this study addresses those challenges. Including references to recent studies from the last five years would also enhance context.

We appreciate this suggestion. We have expanded the Introduction section to provide a better overview of sustained-release ocular implants. In particular, we have included additional references to intravitreal devices such as Ozurdex®, as well as intracameral implants (e.g., Durysta®, iDose®). We now emphasize the main limitations of most intraocular devices, including restricted drug spectrum and duration of action. We also highlight how our study addresses these challenges by developing a biodegradable intracameral implant that simultaneously delivers both DEX and MOX HCl, aiming to improve patient adherence and therapeutic outcomes after cataract surgery.

  1. The figure captions could be more precise regarding the evaluation methods, especially for Figures 2, 5, and 6. Additionally, clarifying methodologies within the text - for example, specifying that the quantification of DEX and MOX HCl release over 10 days was done in vitro as stated in Table 6 - would improve clarity, even if somewhat redundant given the detailed methodology section. For Figure 5, adding method details and decoding channel information in the caption would make the article more accessible and easier to follow.

We thank you for this helpful observation. We have modified the mentioned figure captions to include more precise, easy to follow details. For Figures 2-3, we now indicate evaluation by static light scattering. For Figure 5, the caption has been expanded to specify the use of confocal Raman microscopy, the excitation wavelength (785 nm), and the color coding (DEX = blue, MOX HCl = green, PLGA = red). For Figure 6, we now detail heating rate (10 °C/min) and temperature range (30 – 300 °C). Additionally, we have clarified in the Results section that the quantification of drug release presented in Table 6 was carried out in vitro under sink conditions.

  1. A more careful presentation of figures is recommended: Figures 3 and 6 would benefit from sharper fonts, as in low quality of PDF it is hard to read them; Figure 7 should include labels "A" and "B" on images, images better to be separated by a small gap; and Figure 7 requires a clearer scale bar. Also, ensuring consistent font styles and sizes across all figures would improve visual coherence. While not mandatory for publication nor affecting scientific value, these improvements would enhance the article’s appeal to future readers.

We thank you for these valuable suggestions regarding figure presentation. We have improved the readability of Figures 3 and 6 by adjusting font sizes. Figure 7A and 7B already depict the distinctive labels A and B in the version uploaded separately from the tentative image appearing in the manuscript. The panels are now also separated by a small gap. The scale bar has been reformatted for better visibility.

  1. Although the methods section is clearly structured and sequential, adding introductory and concluding sentences to sections or paragraphs would help readers better understand the authors’ goals, findings, and the significance of results—especially for those less familiar with the topic. The implant synthesis procedure remains somewhat unclear, so a more detailed description is needed, as the current methodology section does not fully explain the process.

We appreciate your suggestion. We have added introductory and concluding sentences to several key parts of the Materials and Methods section to better highlight the rationale and significance of our experiments. In particular, the implant synthesis procedure has been clarified by explicitly describing the choice of hot-melt extrusion as a method to achieve homogeneous drug distribution within the PLGA matrix and providing reasons for the chosen implant dimensions. These several additions should make the methodology clearer and more accessible to readers.

  1.     It would be beneficial to include more SEM images at different magnifications and at various stages of the synthesis process to provide a clearer understanding of the implant development.

We appreciate your suggestion to include additional SEM images. However, the purpose of the SEM analysis in this work was limited to assessing possible morphological changes induced by gamma sterilization, rather than to monitor each step of the manufacturing process. Since implant morphology was not a controlled parameter in our formulation strategy, we did not collect SEM images at different fabrication stages. Therefore, no additional SEM data are available.

  1. If possible, comparing the release rates of similar intraocular delivery systems and discussing what release rate is considered optimal or required would add valuable context.

We appreciate the reviewer’s suggestion. To our knowledge, the only intracameral sustained-release formulation directly comparable to our system is Surodex®, which delivers approximately 60 µg of DEX over 7–10 days. However, detailed release profiles for this product are not publicly available, making a direct quantitative comparison impossible. Other intraocular implants currently on the market are intravitreal (e.g., Ozurdex®) or are designed for the long-term management of open-angle glaucoma (e.g., Durysta® with bimatoprost, ≈ 3–4 months; iDose® with travoprost, up to several years). These systems have very different therapeutic goals, drugs, and time frames, and are therefore not relevant comparators for our intended short-term postoperative application. For this reason, we limited our discussion to Surodex® as the only formulation of similar design and indication, while emphasizing how our implant addresses both inflammation and infection prophylaxis simultaneously. We did however, include a new statement about the performance of our insert compared to what we know about the release rate reported for Surodex®.

  1. How closely do the in vitro release conditions mimic intracameral conditions? Could the protein composition and chemical microenvironment in vivo significantly alter the drug release rate? Also, how quickly does the released drug diffuse under intracameral conditions?

We thank you for this important question. The in vitro release assays aimed to demonstrate that the implants were able to provide sustained-release of both drugs under controlled conditions. As commonly reported in the literature, we used isotonic saline (0.90 % NaCl) to approximate intracameral fluid. We acknowledge this model does not fully replicate the protein composition or biochemical microenvironment of the aqueous humor, which may indeed influence polymer degradation and drug release kinetics. In fact, our in vivo observations indicate that the implants fully disintegrate within ~ 20 days, whereas in vitro only about 25 % of DEX and 45 % of MOX HCl were released after 10 days. This suggests that intracameral conditions accelerate both degradation and diffusion compared to the simplified saline model. For this reason, the in vitro study should be interpreted as a preliminary approximation, while the in vivo results provide a more realistic picture of the implant’s behavior. Further analysis into the pharmacokinetics of both drugs in aqueous humor is currently underway.

Reviewer 2 Report

Comments and Suggestions for Authors
  1. Section 2.14 -lines 216-218- how the size, shape ,position, ocular reactions are measured? Please include a detailed methodology
  2. Section 3.1.1 - figure 11- size of ocular insertion - how it is evaluated?  how the degradation was measured in terms of size of insert? Include a clear methodology
  3. Section 4.5- lines 455-459 - include proper literature support for the observations
  4. Section 4.2- describe clearly whether the size reduction was performed for MOX or not? methodology is not clear please improve it.
  5. Section 4.5- improve the discussion by connecting the formulation factors with drug release. How the kinetics of drug release is affected by physico chemical properties of PLGA, DEX, MOX HCl. Especially when the drugs have different solubility and discussion on their interaction with PLGA and release medium, PLGA degradation mechanism etc. This discussion should be improved.
  6. Table 9 - explain why there is drop in amount of DEX&MOX HCl released between  90 &180 days , 270 and 365 days. is it related to performance of formulation or related to any analytical methodology or analytical error? or drug being fluxed out from the set up during the study? Improve the clarity of discussion and provide supportive literature.
  7. Section 4.11- elaborate the discussion on the effect of formulation factors on degradation of ocular inserts. What is the rationale for the duration of 35 days?

Author Response

  1. Consider briefly mentioning current limitations of marketed products like Surodex, to better position the novelty of the dual-drug implant developed here.

We appreciate this suggestion. We now emphasize the main limitations of most intraocular devices, including restricted drug spectrum and duration of action. We also highlight how our study addresses these challenges by developing a biodegradable intracameral implant that simultaneously delivers DEX and MOX HCl, aiming to improve both patient adherence and therapeutic outcomes after cataract surgery.

  1. In section 2.3, include more specifics on the steel balls (e.g., size, material) used for milling to enhance reproducibility.

We thank you for point out this detail. We have updated the information as requested. A total of 25 stainless-steel balls were used. 5 of them had a diameter of 20 mm, while the rest had a dimeter of 10 mm. The container volume was estimated to be 0.58 L.

  1. In section 2.6, describe the criteria for choosing the implant dimensions (4mm x 0.6mm) and drug loading doses.

We appreciate the comment. We have explained the rationale for choosing these dimensions, which is actually a consensus between tolerability in the anterior chamber and drug loading. Both of them are important parameters for our implants.

  1. For the in vivo studies in section 2.14, clarify if both eyes received implants and if any randomization or blinding was employed.

We thank you for point this out. We have updated the information as part of the ARRIVE guidelines required by the editor. This information also appears now in the manuscript.

  1. It would be helpful for reader if authors can include chromatograms of drug peaks and degradation products in supplementary materials for transparency.

We thank you for this suggestion. All raw data and chromatograms obtained during the analytical validation are available in our records. However, given the already extensive length of the manuscript, we opted not to include these additional figures in order to keep the presentation concise. However, we have already added a statement in the Data Availability section clarifying that chromatograms and related raw data are available from the corresponding author upon reasonable request.

  1. The homogeneity and optimal mixing time determinations are convincing, but supplementing the mixing homogeneity results with more statistical analysis or graphical representation could improve clarity.

We thank you for this comment. The statistical treatment was based on relative standard deviation (RSD) and confidence intervals, which are presented in Table 4, and these parameters already demonstrate the robustness of the mixing process. As the differences between mixing times were small and the values are already statistically meaningful, we believe that additional plots would not substantially increase clarity. For this reason, we preferred to present the data in tabular form, which concisely conveys the key finding that a 120-minute mixing time yielded the most homogeneous blend. It is also worth mentioning that the results shown, come from statistically evaluating a total of 5 points for each of the times studied (60/90/120/150/180 minutes), yielding a total of 25 data points. All this information is also available upon request.

  1. Release profiles (Figures 3A and 3B) are informative and well-explained; however, consider presenting cumulative release versus time graphs alongside to complement drug release kinetics discussions.

We appreciate this positive feedback and suggestion. We would like to clarify that Figures 3A and 3B already depict the cumulative percentage of DEX and MOX HCl released as a function of time, as also detailed in Table 6. Since the current graphs already represent cumulative release profiles, we believe that additional plots would not add further clarity and have therefore opted to keep the current presentation.

  1. The stability study results are intriguing, showing clear differences between accelerated and long-term conditions. Adding discussion on potential mechanisms driving these differences (e.g., polymer degradation, drug-polymer interactions) would strengthen the interpretation.

We thank you for this helpful suggestion. We have expanded our interpretation of the stability study results. We now explain that the accelerated conditions likely promoted hydrolytic degradation of PLGA, leading to increased porosity, which in turn facilitated faster drug release, particularly for DEX. In contrast, MOX HCl was less affected by polymer degradation. We also note that potential permeation of the drug under harsher conditions may have contributed to the altered release rate. These mechanisms help explain the pronounced differences between accelerated and long-term storage.

9 The in vivo rabbit studies are an important preliminary safety and degradation assessment. Consider including more quantitative data for inflammation markers or histopathology to substantiate the absence of toxicity.

We thank you for this valuable suggestion. The in vivo experiments were designed as an exploratory and qualitative assessment of safety and degradation. For this reason, we did not include quantitative markers of toxicity. Instead, the implanted eyes were evaluated clinically by an experienced ophthalmologist, who confirmed absence of toxicity. PLGA and also both DEX and MOX HCl are well-established as non-toxic in intraocular use at even higher doses, which is consistent with our findings.

  1. Expanding on the clinical translation potential and any foreseeable challenges (e.g., implantation technique, implant migration, patient variability) would be beneficial.

We appreciate this comment. We now expand on the clinical potential of the implant, highlighting its ability to deliver both anti-inflammatory and antimicrobial therapy through a single intraoperative administration. We also acknowledge challenges, including implantation logistics, possibility of adhesion or migration within the anterior chamber, and variability in aqueous humor dynamics and implant degradation rate.

  1. The mention of 17-oxodexamethasone degradation product is important; more detail on its potential impact on efficacy and safety could be included.

We thank you for this valuable observation. This degradation product was consistently observed at levels within the tolerance reported for Ozurdex®. At such low levels, it is unlikely to compromise the anti-inflammatory efficacy of DEX. 17-oxodexamethasone is a known degradation product of DEX and to the best of our knowledge no additional toxicological concerns have been reported for it. Therefore, it does not raise significant efficacy or safety concerns in this case. We have, however, added a line stating this.

  1. Some figures (e.g., particle size distributions) would benefit from higher resolution or larger font sizes for axis labels.

We thank you for the suggestion. We have improved the readability of Figures 3 and 6.

  1. Consistency in drug name formatting (DEX vs. dexamethasone, MOX HCl vs. moxifloxacin hydrochloride) throughout the text would enhance readability.

We appreciate this observation regarding consistency. We have updated the manuscript accordingly, and now DEX/MOX HCl are mentioned all across the text.

   14. The inclusion of ethical approval and data availability statements at suitable placer in manuscript is appreciated and aligns with good publication practices.

We thank you for this positive remark.

Reviewer 3 Report

Comments and Suggestions for Authors

The manuscript presents novel and significant findings on a sustained-release intraocular implant delivering dexamethasone and moxifloxacin. Addressing the below points will strengthen the scientific rigor, clarity, and translational relevance of the work.

  1. Consider briefly mentioning current limitations of marketed products like Surodex, to better position the novelty of the dual-drug implant developed here.
  2. In section 2.3, include more specifics on the steel balls (e.g., size, material) used for milling to enhance reproducibility.
  3. In section 2.6, describe the criteria for choosing the implant dimensions (4mm x 0.6mm) and drug loading doses.
  4. For the in vivo studies in section 2.14, clarify if both eyes received implants and if any randomization or blinding was employed.
  5. It would be helpful for reader if authors can include chromatograms of drug peaks and degradation products in supplementary materials for transparency.
  6. The homogeneity and optimal mixing time determinations are convincing, but supplementing the mixing homogeneity results with more statistical analysis or graphical representation could improve clarity.
  7. Release profiles (Figures 3A and 3B) are informative and well-explained; however, consider presenting cumulative release versus time graphs alongside to complement drug release kinetics discussions.
  8. The stability study results are intriguing, showing clear differences between accelerated and long-term conditions. Adding discussion on potential mechanisms driving these differences (e.g., polymer degradation, drug-polymer interactions) would strengthen the interpretation.
  9. The in vivo rabbit studies are an important preliminary safety and degradation assessment. Consider including more quantitative data for inflammation markers or histopathology to substantiate the absence of toxicity.
  10. Expanding on the clinical translation potential and any foreseeable challenges (e.g., implantation technique, implant migration, patient variability) would be beneficial.
  11. The mention of 17-oxodexamethasone degradation product is important; more detail on its potential impact on efficacy and safety could be included.
  12. Some figures (e.g., particle size distributions) would benefit from higher resolution or larger font sizes for axis labels.
  13. Consistency in drug name formatting (DEX vs. dexamethasone, MOX HCl vs. moxifloxacin hydrochloride) throughout the text would enhance readability.
  14. The inclusion of ethical approval and data availability statements at suitable placer in manuscript is appreciated and aligns with good publication practices.

Author Response

  1. Section 2.14 -lines 216-218- how the size, shape, position, ocular reactions are measured? Please include a detailed methodology.

We thank you for this observation. We have now provided a more detailed description of how the implants were assessed in vivo. The cylindrical implants (4 mm × 0.6 mm) were measured with a digital caliper prior to implantation. Their position was monitored by slit-lamp biomicroscopy at postoperative controls. The same method was used to verify shape and integrity. Ocular reactions were evaluated clinically by an experienced ophthalmologist, assessing conjunctival hyperemia, corneal transparency, aqueous flare, keratic precipitates, intraocular pressure, and signs of anterior segment toxicity.

  1. Section 3.1.1 - figure 11- size of ocular insertion - how it is evaluated? how the degradation was measured in terms of size of insert? Include a clear methodology

We thank you for this comment. We have clarified how implant size and degradation were assessed in vivo. The initial dimensions of the implants were confirmed with a digital caliper prior to implantation. After placement in the anterior chamber, size and shape were qualitatively monitored at each postoperative control using slit-lamp biomicroscopy, with serial photographic documentation. All images are available if requested. Degradation was evaluated qualitatively by checking reduction in implant length and integrity until complete disintegration, which occurred within ~20 days.

  1. Section 4.5 - include proper literature support for the observations.

We thank you for this comment. We now mention literature supporting interpretation of the release profiles. We further reference the known effects of PLGA hydrolytic degradation on porosity and water uptake, which contribute to release acceleration.

  1. Section 4.2 - describe clearly whether the size reduction was performed for MOX or not? methodology is not clear please improve it.

We thank you for pointing this out. We have now clarified the size reduction procedure.

  1. Section 4.5 - improve the discussion by connecting the formulation factors with drug release. How the kinetics of drug release is affected by physico chemical properties of PLGA, DEX, MOX HCl. Especially when the drugs have different solubility and discussion on their interaction with PLGA and release medium, PLGA degradation mechanism etc. This discussion should be improved.

We thank you for this insightful comment. We have expanded the text to better connect formulation factors with the observed release kinetics. Specifically, we now discuss how PLGA degradation, aqueous solubility of both drugs and steric factors.

  1. Table 9 - explain why there is drop in amount of DEX&MOX HCl released between 90 &180 days, 270 and 365 days. is it related to performance of formulation or related to any analytical methodology or analytical error? or drug being fluxed out from the set up during the study? Improve the clarity of discussion and provide supportive literature.

We thank appreciate your question. The apparent drops in the 10-day cumulative release values at some long-term time points are actually due to the uncertainty associated with the analytical measurements. In the case of the cumulative release study, five implants from the same batch were simultaneously analyzed to determine a 10-day overall behavior. This was repeated months apart, which means that the exact conditions of the previous sample analysis cannot be exactly replicated (detector performance, standard preparation, environmental fluctuations, possible effects of the storage conditions, etc.). Thus, differences like the ones shown in table 9 are actually expected to happen. It is also worth noting that the confidence intervals we report, overlap each other between sessions. Apart from that, the most logical reasoning would be to think that release rate actually increases with time because of matrix degradation, and not the other way around. This supports the idea that the drops mainly correspond to analytical uncertainty. Considering this as the main cause for the anomalies observed, no additional bibliography would be required. We did, however, clarify in the revised manuscript the number of implants used to assess both cumulative release rate and also uniformity of content throughout the stability study.

  1. Section 4.11- elaborate the discussion on the effect of formulation factors on degradation of ocular inserts. What is the rationale for the duration of 35 days?

We thank you for this insightful comment. With respect to formulation factors, we discuss the degradation process in other sections rather than 4.11, which is intended to explain mainly the biological effects of the implants from a clinical point of view. We have, however, added a few explanations as to why the aqueous humor environment tends to accelerate degradation. On the other hand, we would like to explain that the 35-day observation period was decided for two reasons. Firstly, we wanted to give enough time to the biological system to show signs of possible damage, which fortunately was not observed at all. Secondly, this was in preparation for a follow-up study we are currently working on, involving pharmacokinetic analysis of aqueous humor samples taken at a sufficiently long period of time. We could have also chosen just 25, 30, 40 or 60 days. However, considering the fact that our implants degraded completely after 20 days of exposure to in vivo environment, we believed that 35 days was just sufficient to ensure a proper time window to allow adverse events to happen.

Round 2

Reviewer 1 Report

Comments and Suggestions for Authors

Accept

Reviewer 2 Report

Comments and Suggestions for Authors

Revised version

Reviewer 3 Report

Comments and Suggestions for Authors

The authors have reflected all the said suggestions and comments, which made the manuscript enhanced with improved readability; Thus, I suggest for further consideration with acceptance.